# Imaging of Temporal Bone Mass Lesions: A Pictorial Review

**DOI:** 10.3390/diagnostics13162665

**Published:** 2023-08-13

**Authors:** Marie N. Shimanuki, Takanori Nishiyama, Makoto Hosoya, Takeshi Wakabayashi, Hiroyuki Ozawa, Naoki Oishi

**Affiliations:** Department of Otolaryngology-Head and Neck Surgery, Keio University School of Medicine, 35 Shinanomachi, Shinjuku, Tokyo 160-8582, Japan; nukimari@gmail.com (M.N.S.); tnmailster@gmail.com (T.N.); mhosoya1985@gmail.com (M.H.); t12wakabayashi@yahoo.co.jp (T.W.); ozakkyy123@gmail.com (H.O.)

**Keywords:** flowchart, imaging, pictorial review, temporal bone tumor

## Abstract

Tumoral lesions of the temporal bone include benign or malignant tumors and congenital or inflammatory lesions. Temporal bone lesions are difficult to approach. Therefore, making a preoperative diagnosis and considering whether the lesions require treatment are necessary; if they require treatment, then the type of treatment requires consideration. These tumors cannot be observed directly and must be diagnosed based on symptoms and imaging findings. However, the differentiation of temporal bone lesions is difficult because they are rare and large in variety. In this pictorial review, we divided temporal bone lesions by location such as along the facial nerve, along the internal jugular vein, around the endolymphatic sac, in the internal auditory canal/cerebellopontine angle, petrous apex, middle ear, and mastoid, focusing on the imaging findings of temporal bone lesions. Then, we created a diagnostic flowchart that suggested that the systematic separation of imaging findings is useful for differentiation. Although it is necessary to make comprehensive judgments based on the clinical symptoms, patient background, and imaging findings to diagnose temporal bone mass lesions, capturing imaging features can be a useful differentiation method.

## 1. Introduction

Tumoral lesions of the temporal bone include benign or malignant tumors and congenital or inflammatory lesions. These tumors cannot be directly observed and must be diagnosed based on symptoms and imaging findings. However, the differentiation of temporal bone lesions is difficult. This is because the temporal bone includes various parts from the inner auditory canal to the middle ear and different types of lesions occur in each part. Furthermore, all tumoral lesions are rare. However, among them, several articles have reported lesions with a relatively high frequency in detail [1,2,3] and diagnostic methods have been established. Although studies have summarized the characteristics of each rare tumor, no cross-sectional classification method has been established. Kim et al. evaluated cases of skull-base surgery for intratemporal tumors during the previous 25 years and found that facial nerve schwannomas, squamous cell carcinomas, glomus tumors, and lower cranial nerve schwannomas were the most common [4]. However, other rare tumors have not been reported in detail.

Campion et al. summarized the differential diagnoses of inflammatory lesions in the temporal bones of children [5]. This suggested that a similar flowchart for tumoral lesions and adult cases would be useful for diagnosis.

The temporal bone includes numerous vital structures such as the cochlea, semicircular canal, facial nerve, internal carotid artery, and jugular vein and forms the lateral part of the skull base; therefore, temporal bone lesions are difficult to approach [4]. Making a preoperative diagnosis and considering whether the lesion requires treatment are necessary; if treatment is required, the type of treatment must be determined. The treatment varies, depending on the type of lesion. For example, cholesterol granulomas require a surgical approach that allows drainage, while cholesteatomas require the total removal of the epithelium and a field of view that allows total removal. Not only surgical approaches but also radiological treatments are useful for some lesions.

A differential diagnosis requires a combination of various factors such as age, clinical symptoms, site, and imaging findings. In this pictorial review, we focused on the imaging findings of temporal bone lesions and created a diagnostic flowchart.

A PubMed literature review was performed. Search terms included “temporal bone tumor” and “imaging”. Further literature was searched for each section.

## 2. Review

### 2.1. Along the Facial Nerve

Facial nerve schwannomas, hemangiomas, and the perineural spread of tumors extend along the facial nerve. Facial nerve schwannomas may cause hearing loss and facial nerve paresis. Facial nerve schwannomas exhibit expansile destruction of the surrounding bone on computed tomography (CT) images and a tubular or sausage-link appearance with contrast effects on magnetic resonance imaging (MRI) images [1] (Figure 1). Hemangiomas arise from the perineural venous plexus and occur most often in the region of the geniculate ganglion along the facial nerve [6]. They cause hearing loss and facial nerve paresis. Hemangiomas may show bone spicules within the mass in CT images and MRI typically shows hyperintensity in T1-weighted and T2-weighted images [7]. However, some cases have shown low to isointensity in T1-weighted images, which can make it difficult to distinguish them from facial nerve schwannomas [8]. Moreover, there have been reports of heterogeneous signal intensities and avid enhancements [1]. It is necessary to distinguish between facial nerve schwannomas, hemangiomas, and meningiomas, especially in the geniculate ganglion; however, hemangiomas and meningiomas are differentiated from facial nerve schwannomas by the presence of intertumoral bone spicules [1].

The perineural spread of a malignant tumor is suspected when the thickening and enhancement of the segmental facial nerve are observed. Parotid-gland cancers such as adenoid cystic carcinomas and mucoepidermoid carcinomas as well as nearby skin malignancies that invade or metastasize secondarily to the parotid gland and hematogenous metastases of breast, tracheal, or prostate cancer can all potentially spread along the facial nerve [2].

A comparison of facial nerve schwannomas and the perineural spread of parotid carcinomas showed that facial nerve palsy is less common; target signs are seen in intraparotid facial nerve schwannomas and parotid carcinomas present with an undefined margin [9].

### 2.2. Along the Internal Jugular Vein

Paragangliomas called glomus tumors (glomus jugulare and glomus jugulotympanicum) are most common along the internal jugular vein. Paragangliomas arise from neural crest-derived paraganglia. Paragangliomas extend along the jugular bulb, cranial nerves such as the vagus nerve, the tympanic branch of the glosspharyngeal nerve, and the auricular branch of the vagus nerve. Clinically, dysfunction of cranial nerves IX-XII (in the case of jugular lesions) and pulsatile tinnitus (in the case of tympanic lesions) may occur [6]. Glomus tumors have high vascularity and are homogeneous and significantly enhanced. Bone erosion occurs around the jugular foramen. T2-weighted images show isointensity [2]. Unenhanced T1-weighted and T2-weighted images may show a “salt and pepper” appearance [1] (Figure 2).

Additionally, other lesions extending around the jugular foramen have been reported. Schwannomas of the lower cranial nerves extend along the nerves with moderate to high enhancement. Schwannomas of the lower cranial nerves can be differentiated from glomus tumors by the absence of flow voids. Additionally, meningiomas can occur less frequently. In CT images, meningiomas show sclerosis and hyperostosis with local bone destruction [1]. Using MRI, T1-weighted images show hypointensity to the brain parenchyma and T2-weighted images show isointensity that is often homogenously contrast-enhanced. Cystic degeneration is uncommon and a dural tail sign is a characteristic finding of meningiomas [10]. Meningiomas are differentiated from other lesions by an enhanced dural tail and the absence of flow voids.

The hematogenous metastasis of distant tumors, invasion of nasal carcinomas or chondrosarcomas, and perineural invasion of lymphomas, melanomas, or squamous cell carcinomas may also occur. For patients with a history of cancer or the presence of enhancing and erosive lesions, it is necessary to search for primary cancers [6].

### 2.3. Around the Endolymphatic Sac

Endolymphatic sac tumors and meningiomas are considered to be around the endolymphatic sac. Most endolymphatic sac tumors are sporadic, although some cases happen with Von Hippel–Lindau disease. Endolymphatic sac tumors are locally invasive. The bone invaded by the tumor shows a moth-eaten lytic appearance with intratumoral bone spicules in CT images. Using MRI, smaller tumors show a heterogeneous signal intensity and a peripheral zone of hyperintensity in T1-weighted images; however, the hyperintensity zone of larger tumors is more scattered in T1-weighted images. Enhancement is strong and heterogeneous [1,2]. Glomus tumors are usually located inferior to the labyrinth, whereas endolymphatic sac tumors are located posterior to the labyrinth. However, when tumors are enlarged, it is difficult to differentiate endolymphatic sac tumors from glomus tumors based on their location. Both tumors have high vascularity, but endolymphatic sac tumors usually spare the jugular foramen and do not have a “salt and pepper” appearance [2]. Meningiomas also occur in this area. Tumors occur in the internal auditory canal and cerebellopontine angle. The petrous apex is sometimes expanded in this area.

### 2.4. Internal Auditory Canal/Cerebellopontine Angle

Vestibular schwannomas are the most common, accounting for 60% to 90% of cerebellopontine angle cases. Meningiomas, epidermoids, and nonvestibular posterior fossa schwannomas (trigeminal, facial, and glossopharyngeal nerves) are also common. Additionally, arachnoid cysts, lipomas, dermoids, malignancies (lymphomas, melanomas, and metastases), tumors from the petrous bone (chondrosarcoma), and gliomas have been reported [1].

Patients with vestibular schwannomas experience sensorineural hearing loss, tinnitus, disequilibrium, and decreased speech discrimination. Vestibular schwannomas are isointense or mildly hypointense in T1-weighted images and mildly hyperintense in T2-weighted images with homogeneous contrast. Larger tumors may be heterogeneous and may have cystic components. As described, meningiomas are isointense or slightly hyperintense in T1-weighted images or isointense or hypointense in T2-weighted images and homogeneous. Most vestibular schwannomas grow slowly, from 0.2 mm to a few millimeters per year [1]. Meningiomas seldom expand to the internal auditory canal. The tumor extends along the posterior petrous wall, resulting in an obtuse angle at the bone–tumor interface. In CT images, meningiomas may exhibit calcification, sclerosis, or hyperostosis in the adjacent bone [1]. Epidermoids have hypodensity similar to that of cerebrospinal fluid (CSF) in CT images, thus resembling arachnoid cysts. The smooth remodeling and erosion of petrous bone were observed. Epidermoids show isointensity or mild hyperintensity in T1-weighted images and mild hyperintensity in T2-weighted images. They are characterized by a lack of enhancement and reduced diffusivity in diffusion-weighted images [1]. Dermoids have variable images, depending on the tumor content. MRI shows hyperintensity in T1-weighted images and intense mixing in T2-weighted images. Dermoids often show hyperintensity in diffusion-weighted images [11]. Arachnoid cysts are hyperintense in T2-weighted images with no contrast. Although arachnoid cysts are similar to epidermoids, cholesteatomas, and mucoceles [12], epidermoids and cholesteatomas are hyperintense in diffusion-weighted images and arachnoid cysts are hypointense. Lipomas show fat signals on all sequences such as hyperintensity on nonfat-suppressed T1-weighted images and they may be differentiated from dermoids by a lack of calcification [6].

### 2.5. Petrous Apex

Several studies have described mass lesions in the petrous apex. First, as normal lesions, asymmetric fatty marrow and the effusion of petrous apex air cells may be seen as pseudolesions in the petrous apex. Asymmetric fatty marrow shows hyperintensity in both T1-weighted and T2-weighted images, hypointensity using fat saturation techniques, and fat density using CT [3]. The effusion of petrous apex air cells is hyperintense in T2-weighted images and hypointense in T1-weighted images; however, isointensity or hyperintensity is sometimes observed in T1-weighted images, depending on the protein content. These findings have been observed for cholesterol granulomas and mucoceles as well. CT usually distinguishes between effusion and cholesterol granulomas; effusion shows intact bone margins without expansion and trabecular destruction [3,13].

Mucoceles are rare in the petrous apex. They appear in the pneumatized part of the petrous apex and show bony expansion and trabecular/cortical erosion in CT images. They usually cause no symptoms, but can cause pain or cranial nerve palsy due to the expansion of the bone [14]. Using MRI, they show hyperintensity in T2-weighted images and variable signals in T1-weighted images, depending on the protein content and degree of inspissation of the mucoid material [3]. They have no enhancement or diffusion restrictions. Cholesteatomas, cholesterol granulomas, and trapped effusions may differ [3], but they can be distinguished in T1-weighted and diffusion-weighted images.

Cholesterol granulomas are a foreign-body giant-cell inflammatory reaction to the deposition of cholesterol crystals. They can exist without symptoms, although they present with various symptoms [14]. Cholesterol granulomas present as expansile, sharply defined, and often rounded masses of the petrous apex with cortical thinning and trabecular breakdown in CT images. They show hyperintensity in both T1-weighted and T2-weighted images and a distinct hypointense peripheral rim in T2-weighted images. Smaller lesions are homogenous, whereas larger lesions may be heterogeneous [3] (Figure 3).

Cholesteatomas in the petrous apex show smooth bone expansion and erosion without calcification in CT images. Using MRI, they exhibit hypointensity in T1-weighted images, hyperintensity in T2-weighted images, and isointensity in fluid-attenuated inversion recovery images. Diffusion restriction is characteristic.

Apical petrositis is a rare complication of infectious otomastoiditis that can spread via pneumatized air cells. Initially, only fluid was found in the petrous apex in CT and MRI images. With progressive infections, the invasive destruction of the cortical and trabecular bones is observed [13]. Contrast-enhanced CT shows a heterogeneously enhanced tumor-like appearance; MRI shows fluid of varying consistencies in the middle ear and petrous apex with heterogeneous enhancement. The enhancement and thickening of the adjacent dura may occur [3].

A petrous internal carotid artery aneurysm—a vascular lesion—is rare, but may asymptomatically enlarge. CT shows focal and fusiform dilatation of the bone carotid canal [3]. MRI shows complex or mixed-signal masses in T1-weighted and T2-weighted images and irregular enhancing masses with contrast [3].

Neoplasms in the petrous apex include chondrosarcomas, chordomas, osteosarcomas, meningiomas, myelomas, lymphomas, and metastases. Other regional neoplasms such as trigeminal schwannomas, jugular paragangliomas, and nasopharyngeal carcinomas may also expand to the petrous apex.

Chondrosarcomas tend to initiate from cartilaginous remnants of petrosphenoidal and petrooccipital fissures off the midline [1,13]. They extend laterally and inferiorly and invade the petrous apex and foramen lacerum [1]. CT shows an expansile mass with the destruction of the petrous apex without bone sclerosis. Classically, chondrosarcomas show the so-called arcs and whorls of chondroid matrix calcification [3].

Chordomas arising from the spheno-occipital synchondrosis of the clivus are typically seen on the midline [3,13]. Chordomas are sometimes found off the midline and can resemble chondrosarcomas radiologically [1]. Chordomas invade superiorly or laterally and extend to the petrous apex.

Chondrosarcomas and chordomas are radiologically similar. They may contain calcification and cause the destruction of the surrounding bone in CT images. MRI shows heterogeneous hypointensity in T1-weighted images and hyperintensity in T2-weighted images. Enhancement is prominent but heterogeneous [1]. The differentiation of chondrosarcomas and chordomas at the central skull base is radiologically difficult.

In the temporal bone, hematogenous metastasis most commonly occurs in the petrous apex, accounting for 80% of all cases [15]. Metastasis occurs in breast, lung, prostate, melanoma, and kidney cancer patients [3,13]. The diagnosis is relatively easy when the patient has cancer or other bone metastasis history. However, symptomatic petrous apex involvement rarely occurs in patients with no history of cancer. In such cases, a careful radiological examination is necessary to identify other marrow lesions [3]. It is difficult to distinguish chondrosarcomas, chordomas, invasive or intraosseous meningiomas, and even petrous apicitis, especially in solitary cases. CT shows the insidious infiltration of the marrow space. Symptomatic tumors present as soft-tissue masses with bone destruction varying from sclerotic to permeative. MRI reveals infiltrative enhancing masses. Although the fluorodeoxyglucose uptake varies, positron emission tomography–CT can be useful [3].

Plasmacytomas are solitary tumors and multiple myelomas are characterized by the multifocal proliferation of plasma cells. In CT images, plasmacytomas and multiple myelomas show an expansile, intraosseous, soft-tissue mass with lytic bone destruction. MRI shows isointensity in T1-weighted images, isointensity or hyperintensity in T2-weighted images, and a moderate homogeneous contrast effect in MRI images. Multiple myelomas exhibit other skeletal lesions [3].

### 2.6. Middle Ear and Mastoid

Lesions occurring in the middle ear and mastoid include paragangliomas, facial nerve lesions, hemangiomas, choristomas, hamartomas, meningiomas, perineural spread of tumors, middle-ear adenomas, teratomas, Schneiderian-type papillomas, primary squamous carcinomas, metastases (breast cancer, lung cancer, kidney cancer, prostate cancer, and melanomas), lymphomas, immature teratomas, teratocarcinomas, rhabdomyosarcomas, adenocarcinomas, bone and soft-tissue tumors (Ewing’s sarcoma, fibrosarcoma, osteosarcoma, and chondrosarcoma), osteoblastomas, giant-cell tumors, and Paget’s disease [1,2,16]. Inflammatory lesions such as congenital cholesteatomas, epidermoids, and cholesterol granulomas have also been reported [17]. These lesions are mainly located in the middle ear and mastoid, but sometimes extend to the middle cranial fossa, posterior cranial fossa, or inner ear.

We classified these tumors based on the imaging findings. A few middle ear and mastoid lesions show hyperintensity in T1-weighted images; therefore, T1-weighted images are useful for differentiation. In T1-weighted images, cholesterol granulomas, hemangiomas, and lipomas have been reported as lesions with hyperintensity [1,3]. Cholesterol granulomas usually show heterogeneous hyperintensity in T2-weighted images without enhancement [1]. Hemangiomas may have homogenous hyperintensities in T2-weighted images and avid enhancement [6,7]. However, some studies have reported heterogeneous intensity [1]. Lipomas show homogenous hypointensity in T2-weighted images and fat-suppressed T2 images are considered to be useful.

Many lesions in the middle ear and mastoid show hypointensity in T1-weighted imaging. Among these, hyperintensity in diffusion-weighted images, especially non-echo planar imaging, is characteristic of epidermoids and cholesteatomas [18]. However, diffusion-weighted imaging is not useful for exclusion because it may not detect small lesions. Egmond et al. reviewed the high predictive value of non-echo planar imaging to detect primary and postoperative cholesteatomas [19]. It should be noted that diffusion-weighted imaging also shows hyperintensity in abscesses and inflammatory lesions, which are suspected based on the symptoms, clinical course, and CT findings. Epidermoids also show isointensity in T1-weighted images and mild hyperintensity in T2-weighted images [1] (Figure 4). Cholesteatomas show hypointensity in T1-weighted images and mild hyperintensity in T2-weighted images [1].

We encountered two cases of inflammatory pseudotumors at our hospital and have reported them in detail [20]. For those two cases, both T1-weighted and T2-weighted images had isointensity or hypointensity and homogeneous contrast. Similar cases have been reported [21] but some have presented different signal intensities [20].

Tumors in the middle ear and mastoid with T1 hypointensity and T2 hyperintensity include middle-ear adenomas and hemangiomas. Hemangiomas have been identified as hyperintense lesions in T1-weighted images. However, some reported cases were isointense or hypointense in T1-weighted images, thus causing diagnostic difficulty [8].

Tumors in this area with T1 isointensity or hypointensity and T2 hypointensity include giant-cell tumors, giant-cell reparative granulomas, chondroblastomas, fibrous dysplasia, and neuroendocrine tumors [22,23,24,25]. Giant-cell reparative granulomas have been located in the anterior temporal bone (squamous part); that is, above the temporomandibular joint. Therefore, we considered this location to be a characteristic of giant-cell reparative granulomas [22,26,27,28,29,30]. Giant-cell reparative granulomas have expansile lesions with bone remodeling and thinning of the surrounding bone [23] (Figure 5).

It is difficult to distinguish giant-cell tumors from giant-cell reparative granulomas using a radiological evaluation [22] and a pathological diagnosis is necessary. Giant-cell tumors and chondroblastomas are also difficult to distinguish; therefore, many giant-cell tumors were previously diagnosed as chondroblastomas [23]. Chondroblastomas may appear calcified in CT images [23].

Neuroendocrine tumors are also rare tumors with T1 isointensity and T2 hypointensity [24,25]. These image findings are nonspecific, but Hu et al. reported an unusual new bone formation in neuroendocrine tumors [31]. When bone formation is available in the middle ear, neuroendocrine tumors are included in the differential diagnosis.

Some lesions in the middle ear are detected early because of hearing loss and other symptoms. In such cases, the lesion is small and difficult to detect using MRI [32]. Paragangliomas (glomus tympanicum) are found in the cochlear promontory and are usually small at presentation because they cause otologic concerns such as conductive hearing loss, pulsatile tinnitus, or a retrotympanic mass. Salivary-gland choristomas are rare tumors that appear as soft tissue in the middle ear in CT images, but they are nonspecific. They may be associated with malformations of the ossicles of the ear and abnormal facial nerve aberrations. A pathological diagnosis is necessary [32,33]. Teratomas appear as homogeneous soft tissue in CT images, but may show internal calcification and ossification or a mixture of solid and cystic lesions [16]. Salivary-gland choristomas, teratomas, rhabdomyosarcomas, and Ewing’s sarcomas occur in children; therefore, age is also useful for their differentiation. Metastases from various other malignant tumors may also occur in the middle ear.

## 3. Conclusions and Future Directions

Temporal bone tumors are rare and diverse; therefore, they are difficult to diagnose. Not only imaging findings but also age, clinical symptoms, and site are important for diagnosis. However, imaging findings such as those observed using CT and MRI have a major role in their diagnosis. Therefore, this review focused on the imaging findings. Based on the results, we divided temporal bone mass lesions into several locations and created a flowchart for differential diagnoses using imaging findings (Figure 6 and Figure 7). This flowchart includes all the tumors in this article and suggests that a systematic separation of the imaging findings is useful for differentiation. One limitation of this flowchart is that it does not include all tumors because there is little information available for some rare tumors.

In practice, clinical symptoms may also supplement useful diagnostic information. For example, a history of malignancy suggests metastasis. Rapid progression is more likely an infection or a malignancy. Also, a tumor in the internal auditory canal with facial paralysis raises the possibility of a facial schwannoma. When characteristic symptoms are present, they support reaching a diagnosis in addition to the flowchart.

In imaging diagnosis, artificial intelligence has rapidly advanced in recent years and its usefulness in diagnosing diseases such as chronic otitis media and otosclerosis has been reported [34,35]. It is expected to play an important role in diagnostic imaging in the future. However, temporal bone tumors are rare diseases and it is difficult to accumulate sufficient data for deep learning. Diagnosis by artificial intelligence for rare diseases is still considered to be difficult and the flowchart may be useful for physicians.

Although it is necessary to make comprehensive judgments based on the clinical symptoms, patient background, and imaging findings to diagnose temporal bone mass lesions, capturing imaging features can be a useful differentiation method. Future studies should evaluate the usefulness of this flowchart in clinical practice.

## Figures and Tables

**Figure 1 diagnostics-13-02665-f001:**
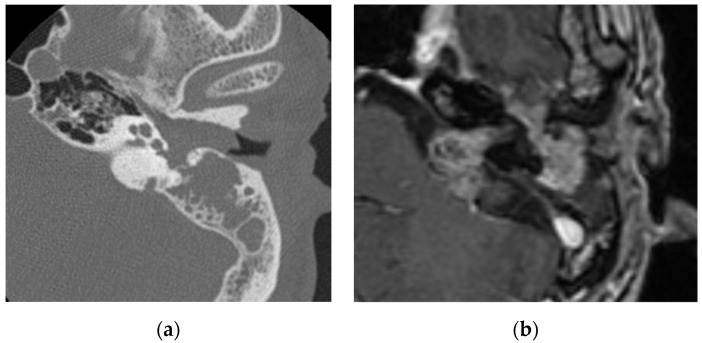
Facial nerve schwannoma: (**a**) in CT, a soft-tissue shadow is seen in the external auditory canal, middle ear, and mastoid with evidence of bone erosion; (**b**) in contrasted T1-weighted images, a mass lesion is visualized in the middle ear, internal auditory canal, and cerebellopontine angle along the facial nerve, which shows enhancement.

**Figure 2 diagnostics-13-02665-f002:**
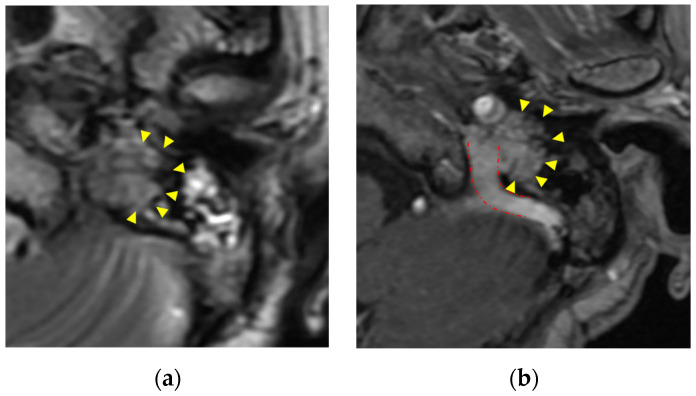
Paraganglioma (glomus jugulare): (**a**) paraganglioma shows moderate intensity with “salt and pepper” appearance in T2-weighted images (yellow triangle); (**b**) homogeneous enhancement in enhanced CT T1-weighted images(yellow triangle). Tumor extends along the internal jugular vein (red dashed line).

**Figure 3 diagnostics-13-02665-f003:**
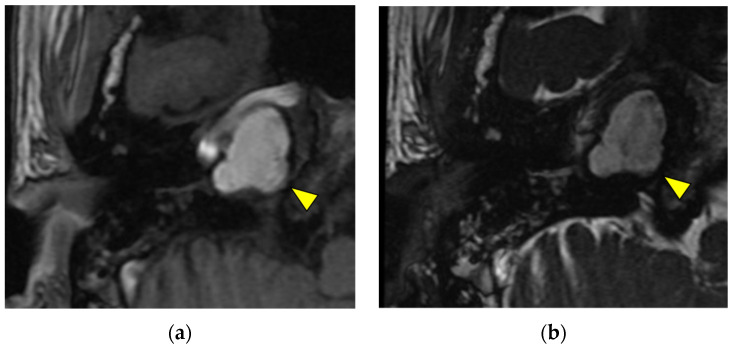
Cholesterol granuloma (yellow triangle) shows hyperintensity in T1-weighted images (**a**) and heterogeneous hyperintensity in T2-weighted images (**b**).

**Figure 4 diagnostics-13-02665-f004:**
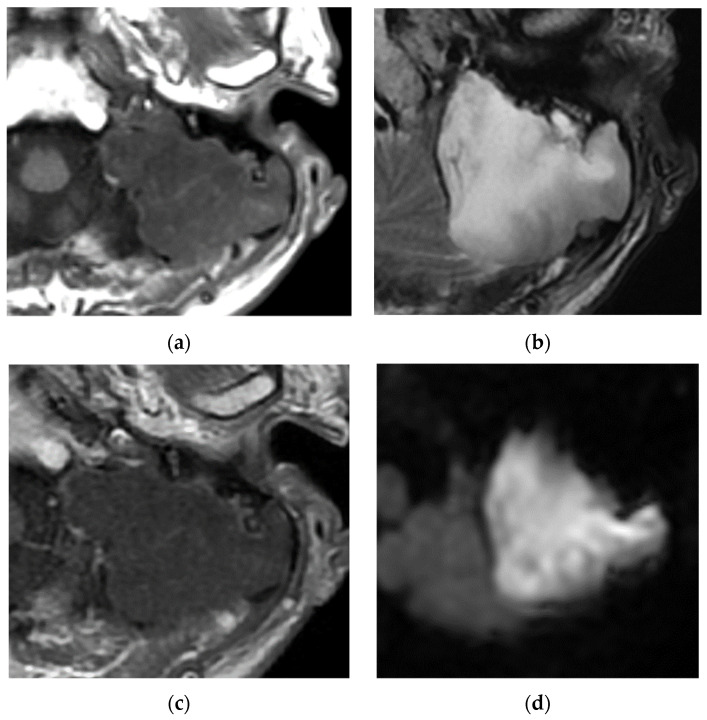
Epidermoids show hypointensity on T1 (**a**) and hyperintensity on T2 (**b**) with no enhancement (**c**). Hyperintensity in DW images is characteristic (**d**).

**Figure 5 diagnostics-13-02665-f005:**
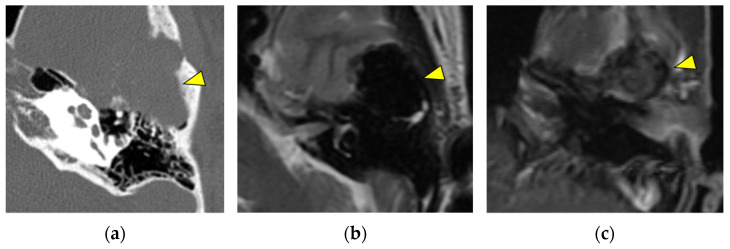
A giant-cell reparative granuloma (yellow triangle), located in the anterior temporal bone, presents as an expansile lesion in CT (**a**). In MRI, it appears hypointense in both T1- and T2-weighted images (**b**) and does not exhibit enhancement in T1-weighted contrast-enhanced images (**c**).

**Figure 6 diagnostics-13-02665-f006:**
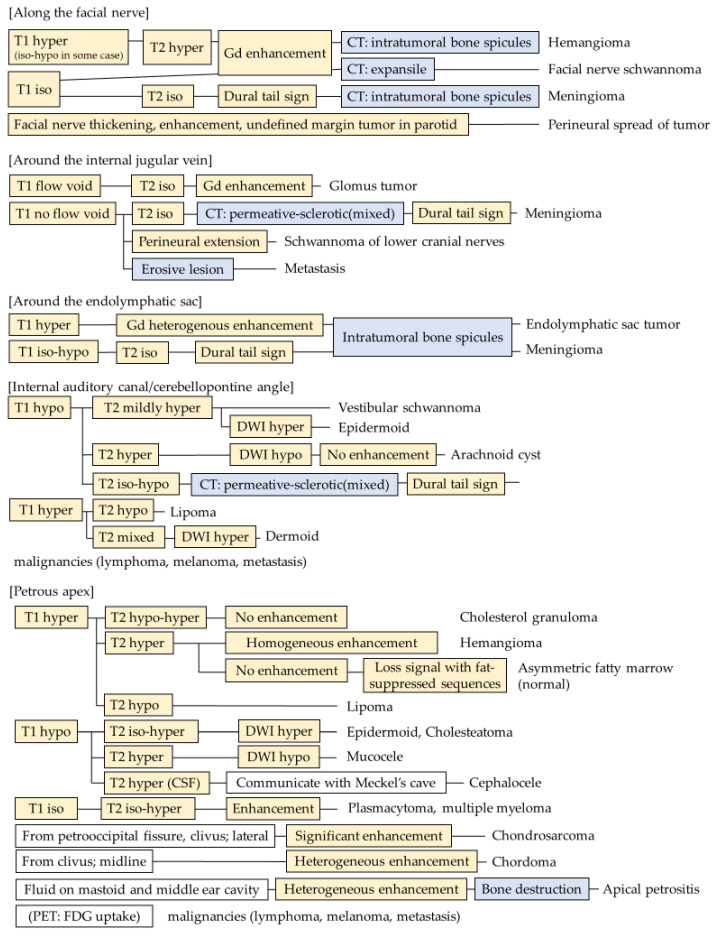
Flowchart for temporal bone tumors at petrous part.

**Figure 7 diagnostics-13-02665-f007:**
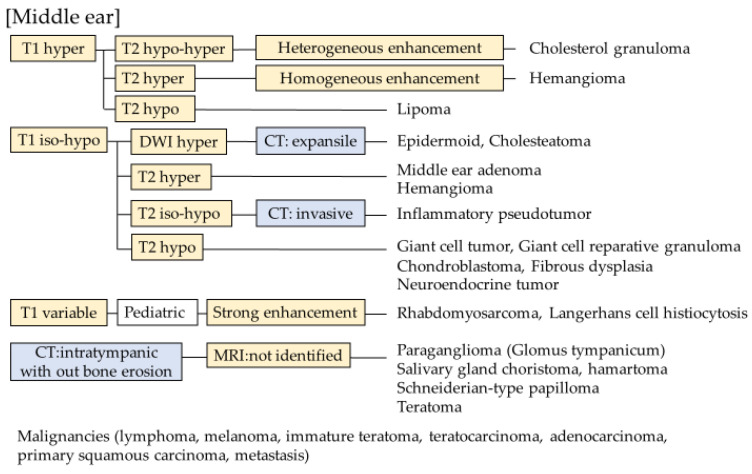
Flowchart for middle-ear tumors.

## Data Availability

Not applicable.

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
