# Peer review of "Imaging of Temporal Bone Mass Lesions: A Pictorial Review"

_diagnostics, 2023, doi:10.3390/diagnostics13162665_

Round 1
Reviewer 1 Report
The current paper is a pictorial review of CT and MRI imaging of temporal bone mass lesions. Due to its specific location, the diagnosis is set through imaging techniques.
After a brief introduction regarding basic data on temporal bone lesions in adults and children, the authors have structured the article using anatomical landmarks.
The first site is the facial nerve trajectory where schannoma, hemangioma and perineural spreading of malignant tumoras along the way are described.
For the jugular vein the authors present paragangliomas, schwannomas and meningiomas. The endolymphatic sac tumors have their particular features presented. For the internal auditory canal and cerebellopontine angleschanommas, meningiomas and epidermoids are featured, while for the petrous apex mucoceles or other fatty structures are presented.
A distinct extended section of the paper regards middle ear mastoid lesions including tumors, both benign and malignant, hemangiomas or inflammatory lesions.
The main contribution of the authors is represented by the two comprehensive yet very clear flow charts regarding MRI features of temporal bone tumors at petrous part and middle ear tumors, as they are extremely helpuful in differentiating between the various pathological entities.
I consider that the paper can be accepted in the current form.
Author Response
Thank you very much for your thoughtful comments. We have modified our manuscript according to the other reviewer's comments.
Reviewer 2 Report
Overall is an interesting article which has achieved his goal for making a preoperative correct diagnosis in temporal bone mass lesions, using a combination between CT and MRI imaging information. Creation of flowcharts including imagistic information for the most frequent types of these lesions, is a very good ideea and added an important contribution for finding the differences between their malignant and benignant imagistic aspects.
I have just a little comment regarding the importance of clinical findings of these lesions, which can be added in some flowcharts and how these flowcharts can be able to replace the human interpretation, if they can, using artificial intelligence, in the future.
Author Response
Thank you very much for your comments. We have modified our manuscript according to your comments.
First, we commented on the clinical findings of some lesions as they can support the diagnosis. Second, we added the comment for artificial intelligence in the conclusion, it is expected to have a significant impact on the future of diagnostic imaging. Those revised parts are highlighted in yellow.